# Light-enabled deracemization of cyclopropanes by Al-salen photocatalysis

Carina Onneken[1,4], Tobias Morack[1,2,4], Julia Soika[1], Olga Sokolova[1], Niklas Niemeyer[1,3], Christian Mück-Lichtenfeld[1,3], Constantin G. Daniliuc[1], Johannes Neugebauer[1,3 ✉] & Ryan Gilmour[1 ✉]

Privileged chiral catalysts—those that share common structural features and are enantioselective across a range of reactions—continue to transform the chemical-research landscape[1]. In recent years, new reactivity modes have been achieved through excited-state catalysis, processes activated by light, but it is unclear if the selectivity of ground-state privileged catalysts can be matched. Although the interception of photogenerated intermediates by ground-state cycles has partially addressed this challenge[2], single, chiral photocatalysts that simultaneously regulate reactivity and selectivity are conspicuously scarce[3]. So far, precision donor–acceptor recognition motifs remain crucial in enantioselective photocatalyst design[4]. Here we show that chiral Al-salen complexes, which have well-defined photophysical properties, can be used for the efficient photochemical deracemization[5] of cyclopropyl ketones (up to 98:2 enantiomeric ratio (e.r.)). Irradiation at $\lambda = 400$ nm (violet light) augments the reactivity of the commercial catalyst to enable reactivity and enantioselectivity to be regulated simultaneously. This circumvents the need for tailored catalyst–substrate recognition motifs. It is predicted that this study will stimulate a re-evaluation of many venerable (ground-state) chiral catalysts in excited-state processes, ultimately leading to the identification of candidates that may be considered 'privileged' in both reactivity models.

Strategically harnessing light as an external stimulus to overcome ground-state reactivity boundaries is a core research endeavour in contemporary catalysis[6,7]. Advances in catalyst design[3,8–10] and streamlined operational platforms[11] have culminated in a diverse arsenal of methods to access excited electronic states through irreversible activation modes. These strategies mitigate microscopic reversibility[12–15] and reduce the dependency on stoichiometric reagents and hazardous operating conditions, thereby allowing reactive species to be generated under mild conditions that are compatible with sensitive environments. Applications in bioconjugation[16] and cellular mapping[17,18] further reflect the breadth and impact that this renaissance continues to enjoy across the scientific landscape. Although this success highlights the effectiveness of photocatalysis in forging new bonds, the non-covalent nature of activation, coupled with the high reactivity of the intermediates that are generated, render enantiocontrol a conspicuous challenge (Fig. 1a). An expansive solution has proved to be dual catalysis[19–21], a regime in which the photocatalyst operates together with an established ground-state chiral catalysis manifold. Enantioselective bond-forming events typically occur from a secondary photoreaction involving a light-generated intermediate. By contrast, processes that use chiral photocatalysts to directly confer enantioselectivity are conspicuously underrepresented: this accentuates the challenges associated with identifying and developing 'privileged chiral photocatalysts'[22–24]. Seminal work by

Bach et al. has established the effectiveness of lactam-based scaffolds, derived from Kemp's triacid, in orchestrating enantioselection in photochemical processes: this blueprint emulates biological recognition in which complementary hydrogen-bonding motifs in the catalyst and substrate ensure structural pre-organization[25]. The modularity of this venerable organocatalyst chromophore can be tuned to enable both enantioselective energy transfer[26] and single-electron transfer[27] processes for substrates bearing a suitable amide group. Substrate-based recognition motifs have also been successfully used in the development of chiral Ir(III) complexes for enantioselective photocatalysis. Elegant studies by Meggers and colleagues have established that acyl imidazole substrates engage with chiral Ir(III) Lewis acids to enable direct, visible-light-induced asymmetric redox catalysis processes[28]. The importance of precision hydrogen-bonding motifs in this context has also been convincingly demonstrated by Yoon, Baik and colleagues to enable enantioselective excited-state photoreactions controlled by a chiral hydrogen-bonding iridium sensitizer[29]. Collectively, these milestones show that remarkable levels of enantioinduction can be achieved under the auspices of a single, chiral photocatalyst when complementary (H-bonding) recognition motifs are present. Expanding this model to include substrates bearing common functional groups for recognition is highly appealing and would ultimately lead to the identification of more general chiral photocatalysts. However, the aim

[1]Institute of Organic Chemistry, Westfälische Wilhelms-Universität (WWU) Münster, Münster, Germany. [2]Department of Chemistry, Yale University, New Haven, CT, USA. [3]Center for Multiscale Theory and Computation, Westfälische Wilhelms-Universität (WWU) Münster, Münster, Germany. [4]These authors contributed equally: Carina Onneken, Tobias Morack. ✉e-mail: j.neugebauer@uni-muenster.de; ryan.gilmour@uni-muenster.de

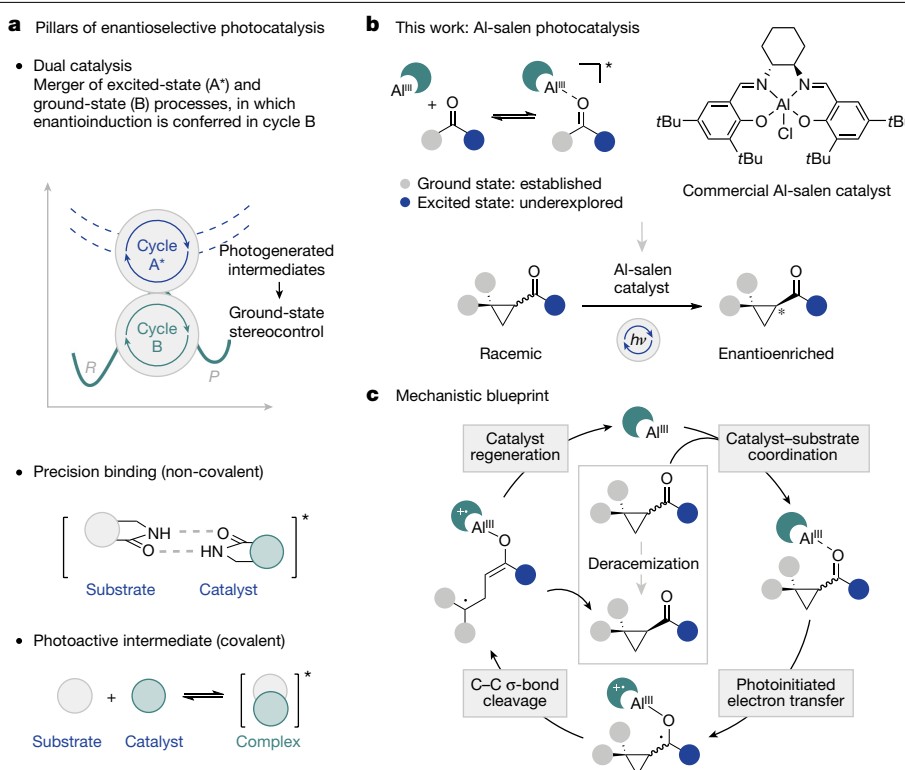

**a** Pillars of enantioselective photocatalysis

- Dual catalysis
  Merger of excited-state (A*) and ground-state (B) processes, in which enantioinduction is conferred in cycle B

- Precision binding (non-covalent)

Substrate    Catalyst

- Photoactive intermediate (covalent)

Substrate    Catalyst    Complex

**b** This work: Al-salen photocatalysis

- Ground state: established
- Excited state: underexplored

Commercial Al-salen catalyst

Racemic    Al-salen catalyst    Enantioenriched

**c** Mechanistic blueprint

Catalyst regeneration    Catalyst–substrate coordination
Deracemization
C–C σ-bond cleavage    Photoinitiated electron transfer

**Fig. 1 | Enantioselective photocatalysis. a**, The pillars of contemporary enantioselective photocatalysis: dual catalysis; non-covalent precision binding and formation of photoactive, covalent intermediates. **b**, Al-salen complexes as privileged catalyst scaffolds in ground-state catalysis and their potential as chiral photocatalysts applied to the deracemization of cyclopropyl ketones. **c**, Design of a mechanistic blueprint that combines the Lewis-acid activation and photoredox properties of Al-salen complexes.

of this endeavour is identifying ground-state recognition modes that can be replicated in an excited-state model[30,31]. This led us to explore the strong oxophilicity of earth-abundant aluminium in creating a Lewis acid–Lewis base component of an emerging chiral blueprint (Fig. 1b). Further confidence in this key interaction stemmed from the well-established photochemistry of aryl ketones, particularly under Lewis-acid activation[32]. Motivated by their success in ground-state catalytic processes, chiral Al-salen catalysts were identified as attractive candidates to regulate primary and secondary photoreactions[33]. A combination of Lewis acidity, a privileged chiral ligand sphere and well-defined optical properties render Al-salen complexes promising candidates to expand the existing chiral photocatalyst portfolio[34,35].

To validate the concept of salen photocatalysis, the deracemization of cyclopropanes was conceived (Fig. 1b): this stemmed from the interest of our group in photocatalytic alkene isomerization[36,37], the reactivity parallels between olefin π-systems and cyclopropyl Walsh orbitals[38], and the historic challenges associated with achieving induction[39,40]. The ability to activate cyclopropanes through adjacent low-lying antibonding orbitals rendered cyclopropyl ketones particularly attractive as substrates for this endeavour[41,42]. The transformation would contribute to the growing interest in light-enabled deracemization of small molecules using low-molecular-weight photocatalysts[43–53]. To this growing repertoire, it was predicted that chiral Al-salen complexes could be effectively used in the deracemization of cyclopropyl ketones through electron transfer to the carbonyl group: this would be achieved through substrate coordination and subsequent excitation of the ligand chromophore, followed by electron transfer in a chiral environment (Fig. 1c).

The foundation of the study rested on combining the LUMO-lowering activation of the aluminium with the strong reducing power of the ligand chromophore in the excited state ($E_{1/2}$(*PC/PC$^+$) ≈ −1.47 V versus saturated calomel electrode (SCE))[54]. It was reasoned that, following substrate coordination to the Al center, excitation of the ligand chromophore would induce electron transfer to generate a transient ketyl radical: this would exist in rapid equilibrium with the achiral, ring-opened form. Relaxation to the ground state by means of back-electron transfer would ultimately regenerate the cyclopropane in the confines of a chiral environment. Accordingly, enantioselectivity would be encoded at the single-chiral-photocatalyst level.

To validate the hypothesis described in Fig. 1, the catalytic deracemization of cyclopropane **rac-1** was investigated using the commercial Al-salen complex **Al-1** in acetone (Fig. 2a) at −70 °C. Guided by the absorption spectra of photocatalyst and substrate (Fig. 2c), selective catalyst excitation was achieved by irradiation of the reaction mixture at 400 nm (3 W light-emitting diode). After 4.5 h, enantioenriched product **(+)-1** was isolated in 78% yield (e.r. 87:13). Encouraged by this result, a process of reaction optimization was initiated (Fig. 2b; for the full optimization table, see the Supplementary Information), beginning with the addition of a soluble salt to stabilize the charge-separated intermediates. The addition of $n$-Bu$_4$NCl enhanced both the yield (80%) and enantioselectivity (e.r. 90:10, entry 1)[55].

Rigorous exclusion of oxygen also proved to be essential in achieving efficient deracemization. The impact of catalyst structural editing was then investigated using a selection of established ligand scaffolds. Notably, commercial catalyst **Al-1** was identified as the optimal catalyst for the transformation of interest. By contrast, the introduction of bulky adamantyl substituents (**Al-2**) proved to be detrimental (e.r. 58:42, entry 2). Moreover, substituting the cyclohexyl backbone with a diphenylethane motif (**Al-3**, entry 3) also resulted in a lower e.r. (80:20) relative to the parent catalyst **Al-1**. Finally, the impact of varying the X-ligand at the Al centre was studied. Although oxygen-bridged dimer **Al-4** was unselective under these reaction conditions (e.r. 48:52, entry 4), replacing the chloro ligand by fluorine preserved efficiency, resulting in only a minor drop in selectivity (e.r. 88:12, entry 5). Altering the

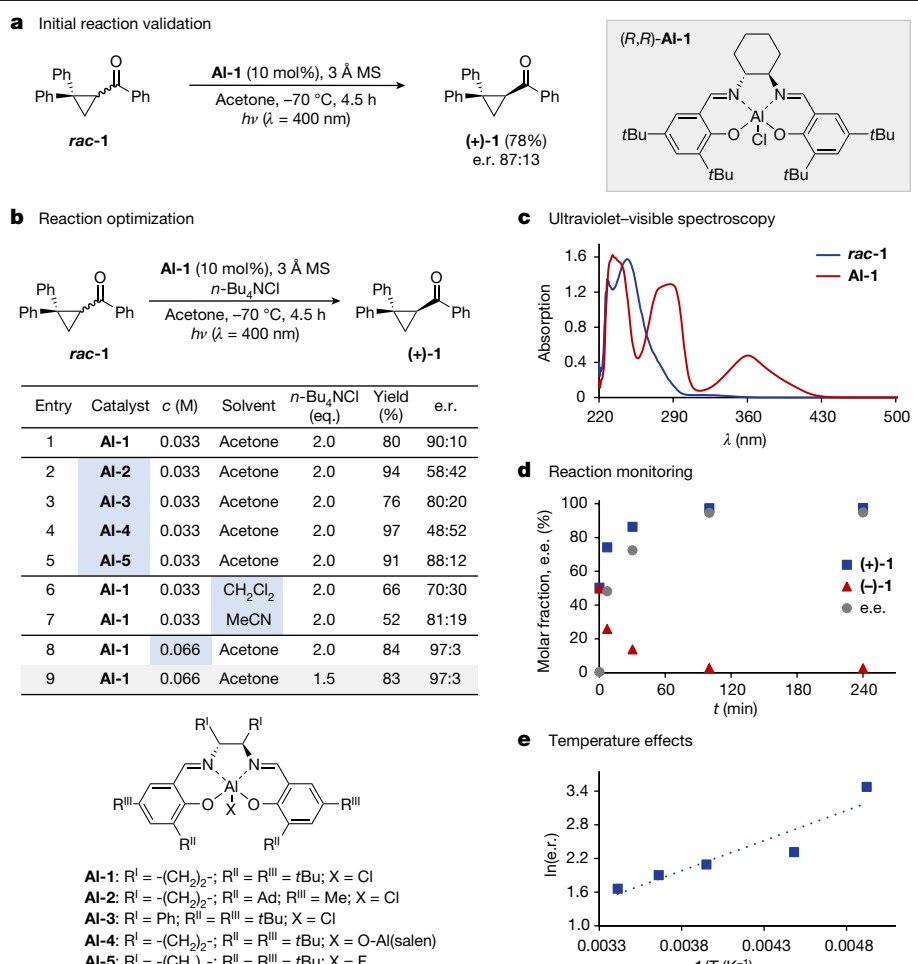

**a**  Initial reaction validation

Al-1 (10 mol%), 3 Å MS
Acetone, –70 °C, 4.5 h
$h\nu$ ($\lambda$ = 400 nm)

rac-1 → (+)-1 (78%)
e.r. 87:13

(R,R)-Al-1

**b**  Reaction optimization

Al-1 (10 mol%), 3 Å MS
$n$-Bu$_4$NCl
Acetone, –70 °C, 4.5 h
$h\nu$ ($\lambda$ = 400 nm)

rac-1 → (+)-1

**c**  Ultraviolet–visible spectroscopy

| Entry | Catalyst | $c$ (M) | Solvent | $n$-Bu$_4$NCl (eq.) | Yield (%) | e.r. |
|-------|----------|---------|---------|---------------------|-----------|------|
| 1 | Al-1 | 0.033 | Acetone | 2.0 | 80 | 90:10 |
| 2 | Al-2 | 0.033 | Acetone | 2.0 | 94 | 58:42 |
| 3 | Al-3 | 0.033 | Acetone | 2.0 | 76 | 80:20 |
| 4 | Al-4 | 0.033 | Acetone | 2.0 | 97 | 48:52 |
| 5 | Al-5 | 0.033 | Acetone | 2.0 | 91 | 88:12 |
| 6 | Al-1 | 0.033 | CH$_2$Cl$_2$ | 2.0 | 66 | 70:30 |
| 7 | Al-1 | 0.033 | MeCN | 2.0 | 52 | 81:19 |
| 8 | Al-1 | 0.066 | Acetone | 2.0 | 84 | 97:3 |
| 9 | Al-1 | 0.066 | Acetone | 1.5 | 83 | 97:3 |

**Al-1**: R$^I$ = -(CH$_2$)$_2$-; R$^{II}$ = R$^{III}$ = $t$Bu; X = Cl
**Al-2**: R$^I$ = -(CH$_2$)$_2$-; R$^{II}$ = Ad; R$^{III}$ = Me; X = Cl
**Al-3**: R$^I$ = Ph; R$^{II}$ = R$^{III}$ = $t$Bu; X = Cl
**Al-4**: R$^I$ = -(CH$_2$)$_2$-; R$^{II}$ = R$^{III}$ = $t$Bu; X = O-Al(salen)
**Al-5**: R$^I$ = -(CH$_2$)$_2$-; R$^{II}$ = R$^{III}$ = $t$Bu; X = F

**d**  Reaction monitoring

(+)-1
(−)-1
e.e.

**e**  Temperature effects

**Fig. 2 | The deracemization of cyclopropyl ketones by means of Al-salen photocatalysis. a**, Initial reaction discovery. **b**, Reaction optimization; reactions performed using **rac-1** (0.10 mmol), Al-salen catalyst (10 mol%), 3 Å mol sieves (MS) (15 mg) and $n$-Bu$_4$NCl in acetone under an argon atmosphere. The reaction mixture was irradiated (400 nm) at −70 °C for 4.5 h using a quartz-glass rod as an optical guiding rod. Yields are reported on the basis of isolated material. The e.r. was determined by high-performance liquid chromatography analysis on a chiral stationary phase. **c**, Absorption spectra of racemic substrate **1** ($c$ = 0.10 mM in CH$_2$Cl$_2$) and chiral Al-salen complex **Al-1** ($c$ = 0.05 mM in CH$_2$Cl$_2$). **d**, Time-course study for the deracemization of **1**. **e**, Effect of varying temperatures on the e.r. e.e., enantiomeric excess.

reaction medium (for example, to CH$_2$Cl$_2$ or MeCN) resulted in lowered reaction efficiency and, therefore, acetone was used for the remainder of this study (entries 6 and 7; for details, see the Supplementary Information). Notably, the reaction showed a marked concentration dependence (entry 8; for details, see the Supplementary Information) and running the reaction at $c$ = 0.066 M was found to be ideal to furnish (+)-**1** in e.r. 97:3 and 84% isolated yield. The loading of the $n$-Bu$_4$Cl additive could also be lowered to 1.5 eq. without any substantial loss to the yield or selectivity (entry 9). Under these optimized conditions, rapid deracemization of **rac-1**, with depletion of (−)-**1**, was observed by reaction monitoring and completion was reached after 100 min (Fig. 2d). An assessment of the effect of temperature on the enantioselectivity revealed a classical temperature dependence in the form of ln(e.r.) ∝ 1/$T$ (Fig. 2e).

Having established optimized conditions, the scope and limitations of this method were investigated (Fig. 3a): this revealed that a variety of modifications could be accommodated. High yields and levels of enantioselectivity were generally observed with *para*-substituted substrates (**1**–**5**, e.r. 90:10 to 98:2) and single-crystal X-ray analysis of enantioenriched **1** enabled the absolute configuration to be assigned as (*S*).

In the solid state, an 84° offset between the planes of *geminal* phenyl groups was observed. To explore if this phenomenon is important in the transfer of chiral information, constrained fluorene cyclopropane **6**

was subjected to the reaction conditions and a notable drop of selectivity (e.r. 70:30) was observed. By contrast, *meta*-substitution was well tolerated, generating **7** in 85% yield and with an e.r. of 90:10. To expand the scope of the method beyond the *geminal* diaryl substituted cyclopropanes, diester substrates **8**–**11** were investigated and found to be highly compatible with the method (up to e.r. 90:10). Finally, the impact of modifying the aryl ketone on the reaction efficiency was explored. Notably, introduction of *para*-substituents was well tolerated, enabling compounds **12** (*p*-F) and **13** (*p*-Cl) to be generated in 80% (**12**) and 89% (**13**) yield and with high levels of enantioinduction (e.r. 94:6 and 88:12, respectively). Reduced levels of selectivity were observed for the *m*-F derivative **14** (91%, e.r. 79:21). To pivot away from *geminal*-substitution, unsymmetrically substituted cyclopropanes were explored within the framework of this salen photocatalysis platform (Fig. 3b). It was tempting to speculate that a photostationary composition consisting of a mixture of enantioenriched diastereomers could be reached through a formal kinetic resolution process. Indeed, irradiation of *geminal* Ph/ester substrates **15** and **16** in the presence of catalyst **Al-1** at low temperatures did generate a photostationary composition, and both diastereomers were generated in almost equal fractions with encouraging levels of optical purity. Subjecting *trans*-**rac-17** to the reaction conditions resulted in formation of the corresponding *cis*-isomer (diastereomeric ratio (d.r.) 45:55 (*trans*:*cis*); e.r. 63:37) and

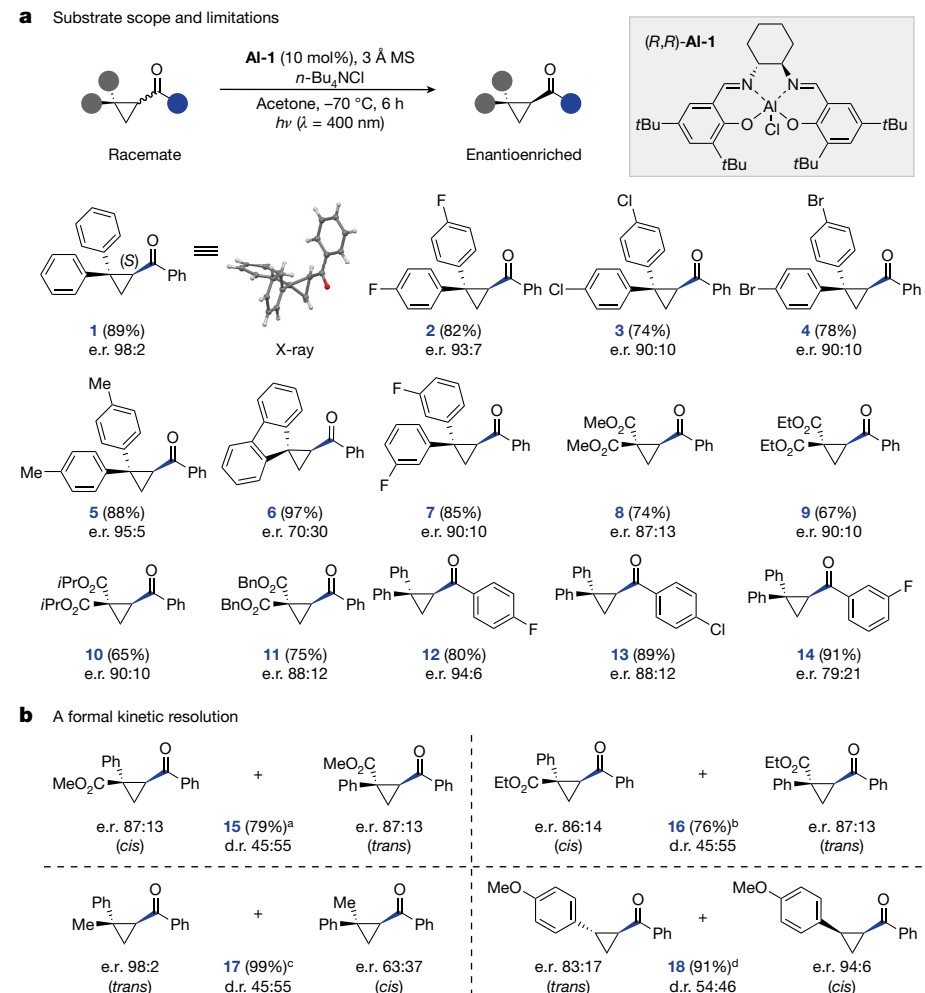

**a** Substrate scope and limitations

**b** A formal kinetic resolution

**Fig. 3 | Investigating the scope of the deracemization of cyclopropyl ketones. a**, Scope of the deracemization; reactions performed using cyclopropyl ketones (0.30 mmol), **Al-1** (10 mol%), 3 Å mol sieves (MS) (45 mg) and $n$-Bu$_4$NCl (1.5 eq.) in acetone (4.5 ml) under an argon atmosphere. The reaction mixture was irradiated (400 nm) at −70 °C for 6 h using a quartz-glass rod as an optical guiding rod. Yields are reported on the basis of isolated material. The e.r. was determined by high-performance liquid chromatography analysis on a chiral stationary phase. **b**, A formal kinetic resolution of unsymmetrically substituted cyclopropyl ketones; reactions performed using cyclopropyl ketones (0.30 mmol), **Al-1** (10 mol%), 3 Å mol sieves (45 mg) and

$n$-Bu$_4$NCl (1.5 eq.) in acetone (4.5 ml) under an argon atmosphere. The reaction mixture was irradiated (400 nm) at −70 °C using a quartz-glass rod as an optical guiding rod. Yields are reported on the basis of isolated material. The e.r. was determined by high-performance liquid chromatography analysis on a chiral stationary phase. [a]Reaction time 4.5 h, starting material d.r. 93:7 (*cis:trans*). [b]Reaction time 4.5 h, starting material d.r. 94:6 (*cis:trans*). [c]Reaction performed on a 0.10 mmol scale, reaction time 65 min (*trans* starting material). [d]Reaction time 64 h using 20 mol% of **Al-1** (*trans* starting material). All experiments were performed in duplicate (see Supplementary Information for the results of single experiments).

(+)-*trans*-**17** in e.r. 98:2 after 65 min. However, on extended irradiation of this reaction mixture, further accumulation of *cis*-**17** with up to e.r. 72:28 was observed (see the Supplementary Information for a more detailed time-course study). Encouraged by these preliminary observations, the behavior of *trans*-di-substituted cyclopropane **18** was investigated. Notably, the enantioselective formation of the *cis*-diastereomer (e.r. 94:6) was accompanied by enantioenrichment of the *trans*-isomer (e.r. 83:17). It is pertinent to note that extended irradiation, and increased catalyst loading, were necessary to reach a photostationary composition.

To place this deracemization enabled by Al-salen photocatalysis on a mechanistic foundation, and provide support for the working hypothesis delineated in Fig. 1, ultraviolet–visible spectroscopic data, reaction-progress monitoring and temperature data were reconciled with experimental observations (see the Supplementary Information for full details). Consistent with a photocatalytic transformation, the deracemization was completely suppressed in the absence of the Al-salen or light: this enabled cyclopropane *rac*-**1** to be recovered

quantitatively (Fig. 4a). Furthermore, omitting molecular sieves compromised reaction efficiency, highlighting the detrimental effect of water on the process. We then turned our attention to the feasibility of the charge-transfer process that is implicit in the mechanism (Fig. 4b). An electron transfer from the excited-state catalyst ($E_{1/2}$(*PC/ PC$^+$) ≈ −1.47 V versus SCE) to substrate *rac*-**1** ($E_{1/2}$ = −1.96 V versus SCE) was considered and, although formally endergonic, it is conceivable in the presence of a Lewis acid[56,57]. Comparison of compound **19** (versus **1**) was intended to demonstrate that the replacement of the *gem*-diPh by *gem*-diEt, although having a marginal impact on the reduction potential, lowers the stability of the radical resulting from C-C bond scission: this was expected to reduce the rate of this step and thus prevent deracemization (e.r. 50:50). This notion is supported by the irreversible reduction of **1** (versus **19**). Replacing the phenyl ketone by a methyl ester, as in the case of compound **20**, results in a more negative reduction potential (<−2.2 V versus SCE), which suggests that the initial photochemical charge transfer from the catalyst is not feasible. Furthermore, control reactions with scalemic **19** and **20** were conducted

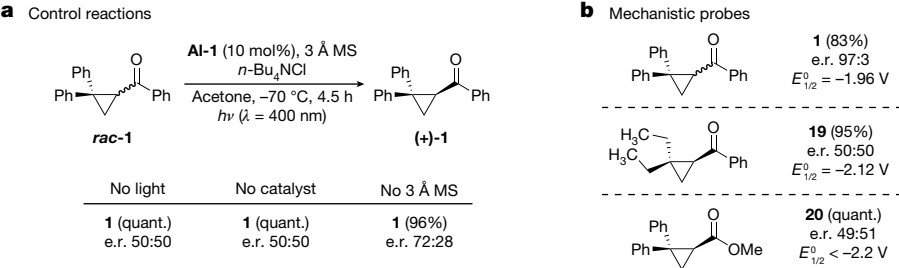

**a** Control reactions

**b** Mechanistic probes

**1** (83%)
e.r. 97:3
$E^0_{1/2} = -1.96$ V

**19** (95%)
e.r. 50:50
$E^0_{1/2} = -2.12$ V

**20** (quant.)
e.r. 49:51
$E^0_{1/2} < -2.2$ V

**Al-1** (10 mol%), 3 Å MS
$n$-Bu$_4$NCl
Acetone, −70 °C, 4.5 h
$h\nu$ ($\lambda$ = 400 nm)

**rac-1** → **(+)-1**

| No light | No catalyst | No 3 Å MS |
|---|---|---|
| **1** (quant.) | **1** (quant.) | **1** (96%) |
| e.r. 50:50 | e.r. 50:50 | e.r. 72:28 |

**c** Computational investigation

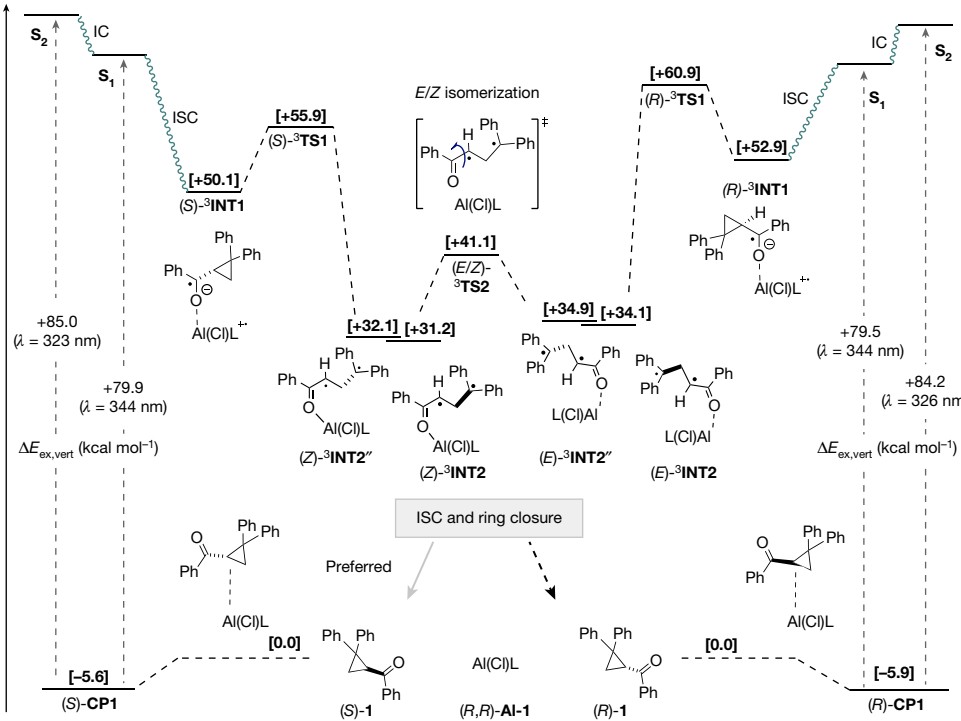

**Fig. 4 | Mechanistic studies. a**, Control reactions; reactions performed on a 0.10 mmol scale. **b**, Cyclic voltammetry and structure–activity relationship studies on the core scaffold. Redox potentials are reported against SCE. **c**, Computational analysis of the reaction mechanism. [$\Delta G_{203}$(acetone)/(kcal mol$^{-1}$)], PW6B95-D3/def2-TZVP + COSMO-RS, $\Delta E_{ex,vert}$ LC-BLYP/def2-TZVP. MS, mol sieves.

and confirmed that no detectable background racemization occurs (for further details see Supplementary Information). To complement these experimental data, a detailed computational study was conducted to further investigate the deracemization mechanism (Fig. 4c).

A conformational analysis of the complex (R/S)-**CP1** formed from cyclopropyl ketone (R/S)-**1** and catalyst (R,R)-**Al-1** revealed non-covalent London dispersion interactions between the aryl rings of the ligand and substrate, but no O–Al bond in the preferred structures. To lend experimental support to this finding, spectroscopic and cyclic voltammetry investigations were conducted and shifts in neither the absorption maxima nor in the cyclic voltammograms were observed (see the Supplementary Information). Therefore, Lewis-acid activation of the substrate in the ground state was discounted, contrary to our initial hypothesis.

Furthermore, no marked energetic discrimination of the two enantiomers of **1** at this complexation stage was noted. With time-dependent density functional theory, very similar vertical excitation energies to the first and second singlet states (S$_1$, S$_2$) of (R/S)-**CP1** were found. Both excited states were characterized by substantial charge transfer from occupied ligand orbitals of the catalyst to the lowest virtual orbitals of the benzoyl moiety in the ketone. To test the assumption that the conformational process required for deracemization occurs in the long-lived triplet state, the charge-separated triplet diradicals

(R/S)-**³INT1** were considered as initial intermediates after intersystem crossing. In contrast to the initial complex, these species exhibit a short Al–O bond in the preferred conformations, as expected for a ketyl radical. The free-energy barrier for ring opening to triplet diradicals **³INT2** is rather low for both configurations (6 and 8 kcal mol$^{-1}$ for (S)-**³INT1** and (R)-**³INT1**, respectively): this step is associated with the back transfer of charge to the metal ligand. During these studies, we did not find notable spin density on the catalyst moiety of **³INT2**. The configuration of the double bond of the enol radical in **³INT2** depends on the cyclopropyl configuration in the low-energy conformation of **³3INT1**: the (Z)-enol radical is formed from (S)-**³INT1**, whereas the ring opening of the (R)-ketyl radical furnishes the (E)-enol radical. Conformational analysis of (E)-/(Z)-**³INT2** reveals that both isomers attain conformations that would yield the (R)-cyclopropane on ring closure, as measured by the dihedral angle around the C˙H–CH$_2$ bond (Supplementary Tables 10 and 11). However, conformations (E)-/(Z)-**³INT2″**, leading to the opposite (S)-stereoisomers, can be found within a free-energy range of 1 kcal mol$^{-1}$ for both diradicals, and epimerization could be easily achieved by rotation of the diphenyl methyl radical group to the opposite side of the enol plane. Furthermore, we have also identified a possible transition structure for the conversion of (E)-**³INT2** to (Z)-**³INT2**, (**³TS2**) with a free-energy barrier of only 7 kcal mol$^{-1}$ for this thermodynamically favorable process. This confirms that comparably

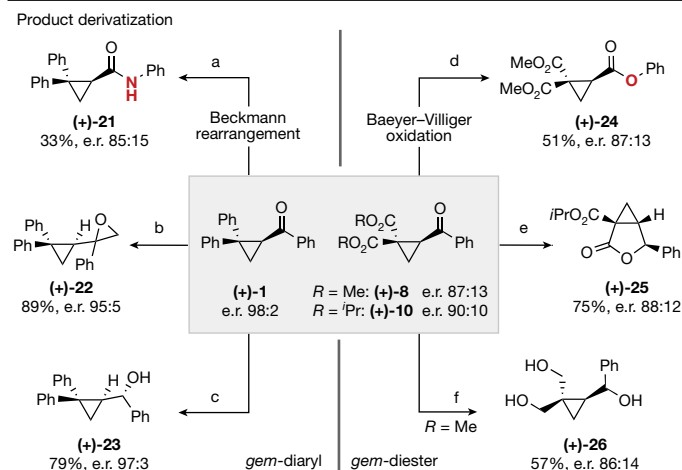

**Product derivatization**

**(+)-21**
33%, e.r. 85:15

a — Beckmann rearrangement

d — Baeyer–Villiger oxidation

**(+)-24**
51%, e.r. 87:13

b

**(+)-22**
89%, e.r. 95:5

**(+)-1**
e.r. 98:2

$R$ = Me: **(+)-8**   e.r. 87:13
$R$ = $^i$Pr: **(+)-10**   e.r. 90:10

e

**(+)-25**
75%, e.r. 88:12

c

**(+)-23**
79%, e.r. 97:3

*gem*-diaryl

f

$R$ = Me

**(+)-26**
57%, e.r. 86:14

*gem*-diester

**Fig. 5 | Functionalization of enantiomerically enriched cyclopropyl ketones following deracemization by Al-salen photocatalysis.** Reaction conditions: (a) 1. Hydroxylamine hydrochloride, pyridine, EtOH, 85 °C, 16 h; 2. Tf₂O, CH₂Cl₂, room temperature (r.t.), 3 h; (b) Me₃SI, NaH, DMSO, r.t., 16 h; (c) LiAlH₄, THF, 0 °C to r.t., 16 h; (d) TFAA, H₂O₂, CH₂Cl₂, 0 °C to r.t., 18 h; (e) LiEt₃BH, THF, 0 °C to r.t., 4 h; (f) LiAlH₄, THF, 0 °C to r.t., 18 h; e.r. determined after derivatization by chiral high-performance liquid chromatography. For full experimental details, see the Supplementary Information.

fast processes can shift the conformational population of triplet diradical (Z)-³**INT2** and that the thermodynamic stability of these intermediates does not necessarily determine the product configuration. Furthermore, we have confirmed by means of spin-flip time-dependent density functional theory calculations (Supplementary Table 14) that (E)-³**INT2** and (Z)-³**INT2** are energetically close to the (open-shell) singlet potential-energy surface, which would lead to facile intersystem crossing in either case. Our computational investigations indicate that there is no substantial stereoselection before and during the excitation process and in the formation of the 1,3-diradical. Fast conformational interconversions of diradicals (E/Z)-³**INT2** can preselect conformations that lead to the observed accumulation of the (S)-cyclopropane during bond formation in the singlet state, assuming that these processes are fast compared with ISC rates.

Finally, to demonstrate the synthetic utility of the method in accessing optically enriched, densely functionalized cyclopropane derivatives, the *gem*-diaryl and *gem*-diester derivatives **(+)-1** and **(+)-8**/**(+)-10** were further modified (Fig. 5). An operationally simple cleavage of the phenyl ketone chromophore was predicted to be advantageous. Therefore, direct conversion to the amide or ester through Beckmann rearrangement[58] or Baeyer–Villiger oxidation[59] was validated. Johnson–Corey–Chaykovsky epoxidation and reduction of **(+)-1**, to generate **(+)-22** and **(+)-23**, proceeded in a fully diastereoselective manner, as did the reduction/cyclization sequence to access the fused lactone derivative **(+)-25** (refs. 60,61). Extensive reduction of **(+)-8** was also efficient and enabled triol **(+)-26** to be prepared without erosion of the optical purity.

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

## Data availability

Data availability CCDC 2217953 contains the supplementary crystallographic data for compound **1**. The data can be obtained free of charge from the Cambridge Crystallographic Data Centre (CCDC; http://www.ccdc.cam.ac.uk/data_request/cif). Supplementary Information is available for this paper. All data are available in the main text or the supplementary materials. Correspondence and requests for materials should be addressed to Prof. Ryan Gilmour (ryan.gilmour@uni-muenster.de).

**Acknowledgements** We gratefully acknowledge the support provided by the technical departments of the Institute of Organic Chemistry at WWU Münster. This manuscript is dedicated with respect and admiration to A. Pfaltz on the occasion of his 75th birthday. Funding: German Research Foundation (SFB 858, IRTG 2678 (GRK 2678-437785492), SFB 1459 (CRC 1459-433682494)) and Fonds der Chemischen Industrie (Kekulé Fellowship to J.S.).

**Author contributions** Initial project idea: T.M., R.G. Conceptualization: C.O., T.M., R.G. Methodology: C.O., T.M., J.S., O.S., N.N., C.M.-L., J.N., R.G. Investigation: C.O., T.M., J.S., O.S., N.N., C.M.-L., C.G.D. Funding acquisition: J.N., R.G. Project administration: J.N., R.G. Supervision: C.M.-L., J.N., R.G. Writing – original draft: C.O., T.M., C.M.-L. Writing – review and editing: C.O., T.M., J.S., C.M.-L., J.N., R.G.

**Funding** Open access funding provided by Westfälische Wilhelms-Universität Münster.

**Competing interests** The authors declare no competing interests.

**Additional information**
**Correspondence and requests for materials** should be addressed to Johannes Neugebauer or Ryan Gilmour.
