## [Peer Review File · Nature]

Manuscript Title: Light Enabled Deracemization of Cyclopropanes by Al Salen Photocatalysis

Reviewer Comments & Author Rebuttals

Reviewer Reports on the Initial Version:

Referees' comments:

Referee #1 (Remarks to the Author):

Overall, the discovery that Salen catalysts can be used to control enantioselectivity in the excited state is of great value, and the transformation described very interesting. The mechanistic investigations, however, seem relatively preliminary and left several unanswered questions:

- In the final sentence of the results section (p12), just prior to the conclusion, the authors propose a hypothesis for enantioenrichment: namely, that it is the conformational dynamics of the acyclic triplet diradical, and its ability to cross to the singlet state in different conformations, that most likely determine deracemization. This seems to be a key point of the study both in terms of how the desymmetrization works and mechanistic novelty, however, there is no experimental or computational evidence that directly addresses this point.

- A series of 3 substrates with different reduction potentials, -1.96 V, -2.12 V and > -2.2V were studied to support the mechanistic proposal, since only compound 1 reacts. However, out of all of these, compound 1 is also the only one that give an irreversible CV, suggesting a bond breaking event (in this case the cyclopropyl C-C). To me this suggests that you need a stabilizing gem-diPh to lower the C-C opening barrier – i.e., reduction may not be what is limiting reactivity of the other two substrates. Also, due to the fact that reduction of 1 is irreversible, the measured redox potential is approximate and very similar to 19. Square-wave voltammetry is necessary to obtain a reliable value. Overall, the experimental data provided to advance mechanistic arguments is limited, consisting of reduction potentials and an absorption spectrum.

- While the Al-Salen is proposed to act as a photoredox catalyst, a simpler explanation might be that it is a bathochromic shift agent. The redox potential of the Al-Salen is measured to result in uphill/endergonic reduction of the substrate, although the authors claim that e-transfer is “conceivable under LA activation of the substrate”. However, in the computational studies, it is discussed that only weakly-bound complexes could be found and that there is no appreciable Al-carbonyl interaction. Doesn't this undermine the suggestion of LA activation?

- The computational protocol looks appropriate for the study, although there is a possibility that only taking the best conformer from CREST and then optimizing with DFT will miss important structures. Are the key comparisons changed quantitatively by taking more xTB conformers forward for DFT optimization?

- The idea that the different diastereomeric complexes give rise to different E- and Z-intermediates is potentially very interesting – although it was not easy to understand based on the shape of the “chiral pocket” why this is the case, even with the current SI Figure. This could be explained in more detail using 3D structures in the SI.

- In Figure 4 the carbon-centered radicals are in the wrong place for the two triplet INT1 species.

Referee #2 (Remarks to the Author):

In this manuscript, Gilmour and coworkers report the deracemization of cyclopropyl ketone substrates using a chiral Al-salen catalyst under 400nm LED irradiation. The reaction scope is limited: cyclopropyl ketone substrates must possess stabilizing diaryl or diester groups. Selectivity is typically moderate-to-good (7:1 to 10:1), although some excellent (49:1) selectivities are observed. The authors nicely anchor scope limitations to mechanistic observations, which is instructive and scholarly.

Despite synthetic limitations, this work is remarkable in several ways.

First, deracemization is a highly significant research challenge with the potential to transform synthetic access to enantioenriched molecules. Very few catalytic deracemization reactions have been reported, and among those sparing examples, most require highly engineered specific substrate-catalyst interactions to achieve high levels of enantioinduction. The only exception – in my opinion – is Luo's approach (ref 51) which allows deracemization alpha to carbonyls. This work advances an Al-salen framework as a chiral photocatalyst, which allows a ketone functional group to be leveraged as a recognition element and thus an important step towards synthetically useful deracemizations.

Second, this study offers a new reactivity framework (C-C bond scission) for deracemization, by breaking/reforming C-C bonds rather than C-H bonds (Knowles, Bach, Hu, etc) or C=C bonds (Bach, Luo). This in principle opens the door for deracemization of fully substituted stereocentres.

Third – and perhaps most importantly – the identification of metal salen complexes as chiral photocatalysts is a potential breakthrough due to their synthetic accessibility and validated stereoselection across reaction platform – hence the authors privileged catalysis pitch.

My reservations about the work are twofold. First, whether the authors' "privileged catalysis" argument can be appropriately validated, as at its core the argument makes claims beyond the present scope of work. Are Al-salen species uniquely viable as photocatalysts? What scope of salen ligands will be tolerated without disrupting desirable photophysical properties? Obviously this may depend on the transformation at hand. However additional screening data would be helpful here – what other salen structures were explored? Is Al required, or would Cr-salen or Mn-salen be effective? Are suboptimal catalysts suffering from poor enantioinduction or perturbation of photophysical properties?

The second concern relates to the mechanism of enantioinduction. DFT study fails to answer most pressing question of why C-C scission is unselective but the *nearly* microscopically identical C-C bond reformation is highly enantioselective. I appreciate that these steps are NOT microscopically identical, but why are the transition state structures for the scission and reformation steps so different? Are the energetics of the two steps very different? Some additional information here is important.

Minor concerns:

The coordination of rac-1 and Al-1 is explored by DFT, but no discussion of an analogous experimental measurement. Is there a ground state interaction detected between substrate and ketone?

It would be useful to run the controls in Scheme 4a with scalemic substrate (instead of racemic) to assess background reactivity (racemization) that cannot be detected using rac-1.

Referee #3 (Remarks to the Author):

The manuscript by Gilmour and Neugebauer et al. describes the visible-light mediated deracemization of cyclopropanes using a bifunctional Al-Salen catalyst. This work represents a conceptually novel approach to asymmetric photocatalysis by combining the LUMO-lowering activation of the ketone as provided by the Lewis acid nature of the Al with the reversible homolytic fission of the cyclopropyl ketone, and asymmetric induction provided therein by the ligand. This manifold is distinct from the chiral H-bonding donor-acceptor energy transfer and ground state chiral LA activation processes.

While this procedure has conceptual merit, the scope and application is somewhat limited by virtue of the reaction design. Further elaboration of the products could be explored, perhaps Baeyer-Villiger oxidation of products could yield the corresponding ester which ultimately could deliver BMS-505130 or Ciprofloxacin via the aldehyde to alkyne etc.

In addition, can the authors comment on the feasibility of performing the deracemization on benzoylcyclobutane, as the B-scission process is viable given appropriate substituents on the ring, although the scission is slower. Moreover, how are substituents at the other cyclopropyl carbon tolerated, such as gem-dimethyl. One could imagine the deracemization of cyclopropanes based upon the Cypermethrin scaffold. If alkenyl substituents are used, can the delocalized radical perhaps close to afford a cyclopentene ring system with asymmetric induction. Additionally, can 3-membered heterocycles undergo similar transformations?

With regards to the mechanism, it would be appropriate to see the UV/Vis and CV of the ligated catalyst and ketone to compare with the free species, this would indicate how much the LA is reducing the reduction potential of the ketone. Is a bathochromic shift in the triplet energy upon coordination of the Al visible at near-IR wavelengths to compare to the computed values? This may also enable some catalyst design. While it is not immediately clear from the images in the SI where the HOMO and LUMO are located in the excited state, if, as perhaps may be expected by analogy to other photocatalysts, the HOMO (and hence reducing ability) resides on the ligand framework, can substitution on the salen Ar rings with EDG's (such as alkoxy or amines) modulate the redox potential such that additional FGs can be tolerated (cmpds. 19 and 30 for instance). Such rational photocatalytic design away from the normal salen ligand would undoubtedly add to this work.

Have the authors looked into the possibility (computationally or mechanistically) of Al-Cl bond

fission, via an LMCT-type process, which would liberate a chlorine radical capable of undergoing HAT. Whether this could lead to enantioenrichment if the LA was bound seems unlikely but is this a competitive pathway energetically? Does the quantum yield of the reaction correlate to the absence of chain processes as purported by the mechanistic studies?

The manuscript is well written and schemes clearly presented, references are adequate. The SI is, again, well-constructed and the purity of compounds satisfactory.

Overall, while I believe this is a conceptual advance in asymmetric photocatalysis, the applicability of this reaction and utility of products remains more modest. If the authors can demonstrate some additional scope (4-membered ring, heterocycle) and/or derivatization to a compound(s) of interest, I believe this is of an appropriate standard to be published in Nature.

Author Rebuttals to Initial Comments:

- **Referee #1 (Remarks to the Author):**

Overall, the discovery that Salen catalysts can be used to control enantioselectivity in the excited state is of great value, and the transformation described very interesting. The mechanistic investigations, however, seem relatively preliminary and left several unanswered questions.

Author comments: We would like to express our thanks to the referee for the generous evaluation of our work on the development of an aluminium salen platform for enantioselective photocatalysis. Over the last months, and following the referee's suggestions, we have conducted detailed experimental and computational investigations to complement the catalysis advances with a deeper mechanistic foundation. We sincerely hope that these address the points raised by the referee (please see below) and fully agree that this elevates the impact of the work.

- In the final sentence of the results section (p12), just prior to the conclusion, the authors propose a hypothesis for enantioenrichment: namely, that it is the conformational dynamics of the acyclic triplet diradical, and its ability to cross to the singlet state in different conformations, that most likely determine deracemization. This seems to be a key point of the study both in terms of how the desymmetrization works and mechanistic novelty, however, there is no experimental or computational evidence that directly addresses this point.

Author comments: This is an excellent, non-trivial aspect of the transformation that the referee has requested be further illuminated. To that end, we have considerably extended the conformer search for the possible triplet structures, including DFT optimisation of more than 20 possible candidate structures per enantiomer. This substantially expands the computational analysis and full details have been added to the supporting information. Figure 4 and the accompanying text in the manuscript have also been altered. To further analyse the origin of enantioenrichment, we have also interrogated the interconversion between the energetically most favourable structures and demonstrated that the isomerisation barrier is small. In addition, we have confirmed (by means of Spin-Flip-TDDFT calculations) that these triplet structures are energetically close to the (open-shell) singlet potential-energy surface, which should lead to comparatively fast intersystem crossing. These results fully support the hypothesis that the conformational dynamics of the acyclic triplet diradical is most likely responsible for the deracemisation.

The manuscript text, supporting Information and Figure 4 have all been modified to accommodate these new findings. The text reads as follows

“To test the assumption that the conformational process required for deracemization occurs in the long-lived triplet state, the charge separated triplet diradicals (R/S)-³INT1 were considered as initial intermediates after intersystem crossing. In contrast to the initial complex, these species exhibit a short Al–O bond in the preferred conformations, as expected for a ketyl radical. The free energy barrier for ring opening to triplet diradicals ³INT2 is rather low for both configurations (6 and 8 kcal/mol for (S)- and (R)-³INT1, respectively): this step is associated with the back transfer of charge to the metal ligand. During these studies, we did not find significant spin density on the catalyst moiety of ³INT2. The configuration of the double bond of the enol radical in ³INT2 depends on the cyclopropyl configuration in the low energy conformation of ³INT1: the (Z)-enol radical is formed from (S)-³INT1, whereas the ring opening of the (R)-ketyl radical furnishes the (E)-enol radical. Conformational analysis of (E)-/(Z)-³INT2 reveals that both isomers attain conformations that would yield the (R) cyclopropane upon ring closure, as measured by the dihedral angle around the C–H–CH₂ bond (Tables S2, S3). However, conformations (E)-/(Z)-³INT2”, leading to the opposite (S) stereoisomers can be found within a free energy range of 1 kcal/mol for both diradicals, and epimerization could easily be achieved by rotation of the diphenyl methyl radical group to the opposite side of the enol plane. Furthermore, we also have identified a possible transition structure for the conversion of (E)- to (Z)-³INT2, (³TS2) with a free energy barrier of only 7 kcal/mol for this thermodynamically favorable process. This confirms that comparably fast processes can shift the conformational population of triplet diradical (Z)-³INT2 and that the thermodynamic stability of these intermediates does not necessarily determine the product configuration. Furthermore, we have confirmed by means of Spin-Flip-TDDFT calculations (Table S6) that (E)- and (Z)-³INT2 are energetically close to the (open-shell) singlet potential-energy surface, which would lead to facile intersystem crossing. Our computational investigations indicate that there is no significant stereoselection before and during the excitation process, and in the formation of the 1,3-diradical. Fast conformational interconversions of diradicals (E/Z)-³INT2 can pre-select conformations which lead to the observed accumulation of the (S)-cyclopropane during bond formation in the singlet state.”

a. Control reactions

No light	No catalyst	No 3 Å MS
1 (quant.) e.r. 50:50	1 (quant.) e.r. 50:50	1 (96%) e.r. 72:28

b. Mechanistic probes

c. Computational investigation

- A series of 3 substrates with different reduction potentials, -1.96 V, -2.12 V and > -2.2V were studied to support the mechanistic proposal, since only compound 1 reacts. However, out of all of these, compound 1 is also the only one that give an irreversible CV, suggesting a bond breaking event (in this case the cyclopropyl C-C). To me this suggests that you need a stabilizing gem-diPh to lower the C-C opening barrier – i.e., reduction may not be what is limiting reactivity of the other two substrates. Also, due to the fact that reduction of 1 is irreversible, the measured redox potential is approximate and very similar to 19. Square-wave voltammetry is necessary to obtain a reliable value. Overall, the experimental data provided to advance mechanistic arguments is limited, consisting of reduction potentials and an absorption spectrum.

Author comments: We appreciate the referee highlighting this aspect of the study and for the very helpful suggestions regarding the mechanistic investigations in a more general sense. The three

substrates **1**, **19**, and **20** were designed to probe the influence of structural modifications on reaction efficiency. In the case of the methyl ester (**20**), our intention was to demonstrate that the phenyl ketone is required for photocatalysis. The inclusion of compound **19** was intended to demonstrate that replacement of the *gem*-diPh by *gem*-diEt only mildly effects the reduction potential but lowers the stability of the radical resulting from C-C bond scission, thereby lowering the rate of this step and preventing deracemization. As highlighted by the referee, this notion is corroborated by the irreversibility of reduction of **1** compared to **19**. In contrast, **20** shows a significantly more negative reduction potential, suggesting that initial photochemical charge transfer from the catalyst is not feasible. We have modified the accompanying text in the manuscript in an attempt to make this more transparent and are grateful to the reviewer for highlighting this potential source of confusion. The referee has suggested that square-wave voltammetry be used to obtain reliable values, but we have been unable to access the electrodes required to perform this specific work. However, we believe that the measured reduction potential of **1** obtained by standard CV is reliable enough to draw the mechanistic conclusions described in the study. We hope that this response fully addresses the referee's comments.

The text in the manuscript now reads as follows *“Comparison of compound **19** (vs **1**) was intended to demonstrate that the replacement of the *gem*-diPh by *gem*-diEt, although having a marginal impact on the reduction potential, lowers the stability of the radical resulting from C-C bond scission: this was expected to lower the rate of this step and thus prevent deracemization (e.r. 50:50). This notion is supported by the irreversible reduction of **1** in contrast to **19**. Replacing the phenyl ketone by a methyl ester, as in the case of compound **20**, results in a more negative reduction potential (<2.2 V), which suggests that the initial photochemical charge transfer from the catalyst is not feasible. Furthermore, control reactions with scalemic **19** and **20** were conducted and confirmed that no detectable background racemization was occurring. To complement these experimental data, a detailed computational study was conducted to further investigate the deracemization mechanism (Fig. 4c).”*

- While the Al-Salen is proposed to act as a photoredox catalyst, a simpler explanation might be that it is a bathochromic shift agent. The redox potential of the Al-Salen is measured to result in uphill/endergonic reduction of the substrate, although the authors claim that e-transfer is “conceivable under LA activation of the substrate”. However, in the computational studies, it is discussed that only weakly-bound complexes could be found and that there is no appreciable Al-carbonyl interaction. Doesn't this undermine the suggestion of LA activation?

Author comments: The notion of aluminum-salen complexes acting as bathochromic shift reagents is intriguing and certainly something that we did consider in light of the elegant work by Profs Bach and Yoon. To further investigate this possibility, we performed additional UV/vis absorption spectroscopy experiments to identify any shifts in the substrate absorption upon catalyst addition. This was also conducted with strongly Lewis acidic Et₂AlCl. These experiments did not reveal any shift in the absorption maxima, which supports the working hypothesis. The experimental details and absorption spectra (shown below) have been added to the Supporting Information. Furthermore, we have modified the manuscript to stress that whilst Lewis acid activation was a consideration in the initial reaction design (akin to ground state activation modes), this was evolved based on DFT insight: this indicates that a van-der-Waals complex (no Al-ketone interaction) in the ground state undergoes charge-transfer upon excitation. The lack of experimentally observed ground-state catalyst-substrate interactions in the UV/vis spectrum is also in line with this mechanistic picture.

The following text has been added to the manuscript:

“To lend experimental support to this finding, spectroscopic and CV investigations were conducted and neither shifts in the absorption maxima nor in the cyclic voltammograms were observed (please see the Supporting Information). Therefore, Lewis acid activation of the substrate in the ground state was discounted, contrary to our initial hypothesis.”

To investigate the possibility of the aluminium-salen complex acting as a bathochromic shift reagent, NMR studies were performed (*vide infra*). The substrate and the catalyst were added in differing ratios to NMR tubes and dissolved in deuterated acetone. No shifts were observed by ^{13}C NMR spectroscopy, which further supports the working hypothesis. We fully agree that this experiment is an important control and thank the referee for suggesting it. We also respectfully direct the referee to the CV measurements in the response to Referee #3, which are consistent with these data.

Expanded spectra

Zoomed in region (measured in acetone-d₆)

- The computational protocol looks appropriate for the study, although there is a possibility that only taking the best conformer from CREST and then optimizing with DFT will miss important structures. Are the key comparisons changed quantitatively by taking more xTB conformers forward for DFT optimization?

Author comments: This is an important point and we appreciate the referee highlighting it. We have performed a systematic investigation of the conformational ensemble of diradical intermediate ³INT2 (both (*E*)- and (*Z*) isomers). With our standard DFT approach, we have optimized ~20 conformers included in an ensemble within an energetic window of 2 kcal/mol (ΔG , PBEh-3c) above the lowest conformer. The implications for the mechanism are presented in Figure 5c and discussed in the section reporting the results of the DFT calculations. We hope that this addresses the referee's comment.

- The idea that the different diastereomeric complexes give rise to different *E*- and *Z*-intermediates is potentially very interesting – although it was not easy to understand based on the shape of the “chiral pocket” why this is the case, even with the current SI Figure. This could be explained in more detail using 3D structures in the SI.

Author comments: This is a very helpful suggestion. Following the referee's advice, we have significantly expanded the computational investigation and modified the supporting information accordingly. From these optimised structures that have been introduced in the revision, the shape of the chiral pocket is now visible and we agree that this provides a helpful structural foundation from which to rationalise the outcome. The ESI now shows the chiral pocket (*S*-product favoured) requested by the referee (please see below).

(*S*)-CP1

(*R*)-CP1

- In Figure 4 the carbon-centered radicals are in the wrong place for the two triplet INT1 species.

Author comments: This is an excellent point. The image has been corrected and we appreciate the helpful comment from the referee.

In closing, we wish to express our gratitude to this referee for investing so much time and effort in providing helpful and constructive feedback. These suggestions have deepened the mechanistic insight and further reinforced the working hypothesis.

- **Referee #2 (Remarks to the Author):**

In this manuscript, Gilmour and coworkers report the deracemization of cyclopropyl ketone substrates using a chiral Al-salen catalyst under 400nm LED irradiation. The reaction scope is limited: cyclopropyl ketone substrates must possess stabilizing diaryl or diester groups. Selectivity is typically moderate-to-good (7:1 to 10:1), although some excellent (49:1) selectivities are observed. The authors nicely anchor scope limitations to mechanistic observations, which is instructive and scholarly. Despite synthetic limitations, this work is remarkable in several ways.

First, deracemization is a highly significant research challenge with the potential to transform synthetic access to enantioenriched molecules. Very few catalytic deracemization reactions have been reported, and among those sparing examples, most require highly engineered specific substrate-catalyst interactions to achieve high levels of enantioinduction. The only exception – in my opinion – is Luo's approach (ref 51) which allows deracemization alpha to carbonyls. This work advances an Al-salen framework as a chiral photocatalyst, which allows a ketone functional group to be leveraged as a recognition element and thus an important step towards synthetically useful deracemizations.

Second, this study offers a new reactivity framework (C-C bond scission) for deracemization, by breaking/reforming C-C bonds rather than C-H bonds (Knowles, Bach, Hu, etc) or C=C bonds (Bach, Luo). This in principle opens the door for deracemization of fully substituted stereocentres.

Third – and perhaps most importantly – the identification of metal salen complexes as chiral photocatalysts is a potential breakthrough due to their synthetic accessibility and validated stereoinduction across reaction platform – hence the authors privileged catalysis pitch.

Author comments: We greatly appreciate the generous evaluation from referee 2 and, in particular, for the summary of how the work complements the other conceptual pillars of asymmetric photocatalysis. As the referee notes in point 2, this work provides a new chiral photocatalyst platform for the deracemization via reversible C-C bond scission. In the manuscript, we have highlighted the value of so-called privileged catalysts in ground state processes and questioned if a comparable suite of privileged photocatalysts will emerge in the future. The referee touched on this in her/his/their report, and this motivated us to demonstrate proof of concept in a second, non-related transformation (please see below). We are most grateful to the referee for encouraging us to develop this further.

My reservations about the work are twofold. First, whether the authors' "privileged catalysis" argument can be appropriately validated, as at its core the argument makes claims beyond the present scope of work. Are Al-salen species uniquely viable as photocatalysts? What scope of salen ligands will be tolerated without disrupting desirable photophysical properties? Obviously this may depend on the transformation at hand. However additional screening data would be helpful here – what other salen

structures were explored? Is Al required, or would Cr-salen or Mn-salen be effective? Are suboptimal catalysts suffering from poor enantioinduction or perturbation of photophysical properties?

Author comments: This is a great point and we appreciate the opportunity to comment and revise the manuscript. Salen complexes have found broad application in asymmetric synthesis in ground state regimes and they are often referred to as privileged chiral catalysts. When developing this project, we were very conscious of the value that a toolkit of privileged chiral photocatalysts would confer to complete current strategies. The manuscript was intended to phrase any possible generality with salen-photocatalysis as more of a question than a claim, and to extend the generality of Al-salen complexes from the ground state to excited state reactivity paradigms. However, we fully appreciate that the term “privileged” should be used with extremely caution and we apologise for any lack of clarity. We have identified all incidences where the term appears in the manuscript and modified the text where necessary to ensure that our intentions are clear. We appreciate the referee highlighting this important point.

We are confident that Al-salen photocatalysis will find other applications and, motivated by the referee’s suggestion, we are happy to share a preliminary validation of an enantioselective 6 π -electrocyclization that occurs at ambient temperature (up to 84:16 *er*. Please see below). Although this is beyond the scope of this study on cyclopropane deracemisation, we fully agree with the reviewer that an additional example would strengthen the manuscript and hint towards generality. We have therefore added these data to the supporting information and included a line in the main text of the manuscript. Since the focus of the paper is deracemization, we feel that the work is best placed in the Supporting Information but we would be happy to move it to Figure 5 if that is preferable. The following statement has been added to the manuscript as reference 62.

“To further demonstrate the synthetic potential of Al-salen photocatalysis, catalyst **Al-1** has been utilized in an enantioselective 6 π -electrocyclization. Please see the supporting information for full details.”

The referee has also raised an excellent point regarding the impact of structural alterations on catalyst performance. Early on in the project, and in addition to our original screening data, we explored a set of alternative salen complexes (please see below). The non-redox active metal aluminium is unique in its ability to generate an active chiral salen photocatalyst, whereas other metal-based salens were unreactive. This observation is in line with earlier studies on the photophysical properties of various metal-salen complexes (refs 34 & 35). It is interesting to note that substitution of the *t*Bu group results in lower levels of enantioinduction. These data have been added to the supporting information and we apologise for not including them in the original submission.

We also respectfully direct Referee #2 to the next report where we detail the impact of structural modifications around the salen periphery. Ligands bearing only a methoxy group in the *ortho* or *para* positions were unfortunately insoluble in an array of common solvents (such as DCM, acetone or toluene). We therefore did not explore their photocatalysis behavior further due to this limitation. Substituting the *p*-*t*Bu group with *p*-OMe did provide a catalyst that was soluble but no enantio-enrichment was observed under the standard reaction conditions. The cyclopropane was recovered in racemic form. We consider that the success of the title reaction with a commercial catalyst will lower the barrier to utilising the methodology.

Catalyst central ion screening

entry	catalyst	isolated yield [%]	e.r.
1	1	83	97/3
2	7	94	50/50
3	8	97	50/50
4	9	quant.	50/50
5	10	98	50/50
6	11	97	50/50

Reactions performed on a 0.1 mmol scale according to General Procedure E.

The second concern relates to the mechanism of enantioinduction. DFT study fails to answer most pressing question of why C-C scission is unselective but the *nearly* microscopically identical C-C bond reformation is highly enantioselective. I appreciate that these steps are NOT microscopically identical, but why are the transition state structures for the scission and reformation steps so different? Are the energetics of the two steps very different? Some additional information here is important.

Author comments: We thank the referee for this very helpful and insightful question. This question was also raised by the first referee and therefore have conducted a far more in-depth computational analysis of the reaction mechanism. This was very involved and we appreciate the opportunity to further refine the mechanistic proposal. We respectfully refer the referee to the comments to the first report. By interrogating the interconversion between the energetically most favourable structures it has been possible to demonstrate that the isomerisation barrier is small. In addition, we have

confirmed (by means of Spin-Flip-TDDFT calculations) that these triplet structures are energetically close to the (open-shell) singlet potential-energy surface, which should lead to comparatively fast intersystem crossing. These results fully support the hypothesis that the conformational dynamics of the acyclic triplet diradical is most likely responsible for the deracemization. The manuscript text, associated figure and supporting information have been significantly expanded to address this point.

Minor concerns:

The coordination of *rac*-1 and **Al**-1 is explored by DFT, but no discussion of an analogous experimental measurement. Is there a ground state interaction detected between substrate and ketone?

Author comments: We very much appreciate the referee's suggestion to provide additional data regarding a ground state interaction between *rac*-1 and **Al**-1. We have performed a series of UV-vis absorption spectroscopy experiments to exclude a potential shift of the substrate absorption band upon catalyst coordination. However, no observable shift was detected, which is fully in line with the identification of a van-der-Waals complex (no Al-O interaction) in the ground state by DFT. This is further supported by CV and NMR experiments (please see the responses to referees #1 and #3). Furthermore, addition of the non-absorbing, strongly Lewis acidic Et₂AlCl also failed to induce a detectable shift of the substrate absorption maximum. This additional data has been added to the Supporting Information.

It would be useful to run the controls in Scheme 4a with scalemic substrate (instead of racemic) to assess background reactivity (racemization) that cannot be detected using *rac*-1

Author comments: Assessing background reactivity with scalemic substrates is an excellent idea and we would like to thank the referee for this suggestion. We can confirm that this was not observed and these data have been added to the manuscript and Supporting Information. The following text has been added to the manuscript "*Furthermore, control reactions with scalemic 19 and 20 were conducted and confirmed that no detectable background racemization was occurring*".

In addition, we have included a series of product derivatisation protocols that enable the enantioenriched cyclopropyl ketones to be processed to esters and amides (Baeyer-Villiger and Beckmann rearrangement, respectively) and reduced or cyclised. Please see the response below for full details on the derivatisation reactions.

In closing, we wish to thank the referee for the numerous insightful comments on how to strengthen the manuscript. We fully agree that the additional insight, and the 6 π electrocyclization, elevate the paper.

- **Referee #3 (Remarks to the Author):**

The manuscript by Gilmour and Neugebauer et al. describes the visible-light mediated deracemization of cyclopropanes using a bifunctional Al-Salen catalyst. This work represents a conceptually novel approach to asymmetric photocatalysis by combining the LUMO-lowering activation of the ketone as provided by the Lewis acid nature of the Al with the reversible homolytic fission of the cyclopropyl ketone, and asymmetric induction provided therein by the ligand. This manifold is distinct from the chiral H-bonding donor-acceptor energy transfer and ground state chiral LA activation processes.

Author comments: We appreciate the very kind commentary regarding the conceptual advance disclosed in our study.

While this procedure has conceptual merit, the scope and application is somewhat limited by virtue of the reaction design. Further elaboration of the products could be explored, perhaps Baeyer-Villiger oxidation of products could yield the corresponding ester which ultimately could deliver BMS-505130 or Cipralisant via the aldehyde to alkyne etc. In addition, can the authors comment on the feasibility of performing the deracemization on benzoylcyclobutane, as the B-scission process is viable given appropriate substituents on the ring, although the scission is slower. Moreover, how are substituents at the other cyclopropyl carbon tolerated, such as gem-dimethyl. One could imagine the deracemization of cyclopropanes based upon the Cypermethrin scaffold. If alkenyl substituents are used, can the delocalized radical perhaps close to afford a cyclopentene ring system with asymmetric induction. Additionally, can 3-membered heterocycles undergo similar transformations?

Author comments: We really appreciate the referee's very helpful suggestion to demonstrate that the products can be easily elaborated. Driven by the biological significance of the products, the referee specifically encouraged us to explore Baeyer-Villiger oxidation as a means to remove the phenyl ketone chromophore that is necessary for catalysis. We have developed conditions that enable the ketone to be processed to the ester with no erosion of optical purity. Furthermore, we have included a Beckmann rearrangement protocol to access the corresponding amide directly. Ketone reduction and epoxidation was found to be highly diastereoselective (>20:1), and with the gem-diester derivatives it was possible to access the triol by reduction and generate fused cyclopropyl lactones. This latter class of materials are often generated by biocatalytic cyclopropanation (*Nat. Catal.* **2**, 471 (2019); *J. Am. Chem. Soc.* **141**, 9145 (2019)) and are valuable precursors in bioactive small molecule discovery. The following figure has been added to the manuscript and we thank the referee for the helpful suggestions.

We have systematically investigated the additional suggestions from referee #3 and below is a summary of our findings. The deracemisation of epoxides and aziridines was attempted but was found to be incompatible with the reaction conditions. Additional substituents around the cyclopropane resulted in only low levels of enantioselectivity, and this is likely due to a disruption of decisive catalyst / substrate interactions. In the case of the heterocycles that the referee specifically requested

(epoxides and aziridines), we also explored the impact of the relative stereochemistry on the reaction outcome.

The suggestions regarding substrate classes are very exciting and will certainly drive us to develop solutions in the future and we appreciate the referee highlighting them. We hope that this additional synthetic work fully addresses the question.

Unsuccessful substrates

Epoxide

Aziridine

With regards to the mechanism, it would be appropriate to see the UV/Vis and CV of the ligated catalyst and ketone to compare with the free species, this would indicate how much the LA is reducing the reduction potential of the ketone. Is a bathochromic shift in the triplet energy upon coordination of the Al visible at near-IR wavelengths to compare to the computed values? This may also enable some catalyst design. While it is not immediately clear from the images in the SI where the HOMO and LUMO are located in the excited state, if, as perhaps may be expected by analogy to other photocatalysts, the HOMO (and hence reducing ability) resides on the ligand framework, can substitution on the salen Ar rings with EDG's (such as alkoxy or amines) modulate the redox potential such that additional FGs can be tolerated (cmpds. 19 and 30 for instance). Such rational photocatalytic design away from the normal salen ligand would undoubtedly add to this work.

Author comments: During the process of reaction optimisation, we were fortunate to discover that the commercial Al-salen used was ideally suited to the reaction under investigation. We consider this success with an *off-the-shelf* catalyst to be advantageous in that it allows practitioners to directly utilise the methodology without having to first prepare the catalyst. However, we fully agree that such

modification will play an important part in future studies to expand salen photocatalysis. In the specific cases mentioned by the referee, we have prepared a number of analogues and explored their suitability. Although some of this data was presented in the supporting information, we fully appreciate that more information would be beneficial to readers.

We have performed a range of UV/vis and NMR investigations to probe for possible (Lewis acid) catalyst - substrate interactions in the ground state but such phenomena were never detected (please also see the response to Referee #2 above). This is fully consistent with the DFT analysis. Furthermore, we performed CV measurements to further investigate catalyst and ligand interactions as proposed. However, the shift of the substrate's reduction potential upon catalyst addition in the CV is negligible.

Regarding the helpful suggestion to investigate more electron-rich ligand frameworks, we have prepared the ligands bearing a methoxy group in the *ortho* or *para* positions. Unfortunately, these catalysts were insoluble in an array of common solvents (such as DCM, acetone or toluene). We therefore did not explore their photocatalysis behavior further due to this limitation. Substituting the the *p*-*t*Bu group with *p*-OMe did provide a catalyst that was soluble but no enantio-enrichment was observed under the standard reaction conditions. The cyclopropane was recovered in racemic form. We have included this information (summarised below) in the supporting information.

Modification of the Al(salen) complexes with more electron rich ligands

insoluble in acetone, DCM, toluene

insoluble in acetone, DCM, toluene

Tried under photochemical conditions

Catalyst screening added to the supporting information:

Table S1: Reaction optimization.

entry	solvent	c [M]	catalyst	Bu ₄ NCl loading	isolated yield [%]	e.r.
1	acetone	0.033	1	2 eq.	80	90/10
2	acetone	0.033	2	2 eq.	94	58/42
3	acetone	0.033	3	2 eq.	76	80/20
4	acetone	0.033	4	2 eq.	97	48/52
5	acetone	0.033	5	2 eq.	91	88/12
6	acetone	0.033	6	2 eq.	98	50/50

Reactions performed on a 0.1 mmol scale according to General Procedure E.

Have the authors looked into the possibility (computationally or mechanistically) of Al-Cl bond fission, via an LMCT-type process, which would liberate a chlorine radical capable of undergoing HAT. Whether this could lead to enantioenrichment if the LA was bound seems unlikely but is this a competitive pathway energetically? Does the quantum yield of the reaction correlate to the absence of chain processes as purported by the mechanistic studies?

Author comments: The idea of Al-Cl cleavage is intriguing and we certainly considered this possibility and investigated it computationally. Whilst certainly conceivable, and previously suggested (ref. Baleizão, C., Gigante, B., Ramôa Ribeiro, F., Ferrer, B., Palomares, E. & Garcia, H. *Photochem. Photobiol. Sci.* **2**, 386–392 (2003)), we could not find any evidence for a competitive mechanistic pathway that

involves Cl radicals by DFT. Additionally, **cat. 5** (exhibiting a stronger Al-F bond) is highly effective in the transformation, indicating that Al-Cl bond fission is unlikely to be a productive mechanistic pathway contributing to the efficiency of the deracemization process.

The manuscript is well written and schemes clearly presented, references are adequate. The SI is, again, well-constructed and the purity of compounds satisfactory.

Author comments: We are most grateful to the referee for carefully assessing the manuscript and supporting information in detail, and appreciate this generous evaluation.

Overall, while I believe this is a conceptual advance in asymmetric photocatalysis, the applicability of this reaction and utility of products remains more modest. If the authors can demonstrate some additional scope (4-membered ring, heterocycle) and/or derivatization to a compound(s) of interest, I believe this is of an appropriate standard to be published in Nature.

Author comments: We very much appreciate the generous description of the work as a conceptual advance in asymmetric photochemistry. The suggestions to include derivatisation protocols that would enable the enantioenriched products to be easily process further are extremely helpful and we thank the referee for his/her/thier suggestions. A new figure has been introduced at the end of the manuscript that shows examples of facile oxidation, reduction and cyclisation events. Most importantly, we have established that the phenyl ketone chromophore that is necessary can be derivatised by Baeyer-Villiger oxidation and Beckmann rearrangement chemistry to access the corresponding ester and amide, respectively (*vide infra*). Moreover, we have included preliminary validation of a second, very different transformation enabled by Al-salen photocatalysis. As described above, an enantioselective 6 π electrocyclization has been achieved at room temperature with encouraging levels of enantioselectivity (please see the response to referee #2). We sincerely hope that our responses and the additional data address the concerns of the reviewer and appreciate the numerous helpful suggestions on how to elevate the study.

Fig. 5. Functionalization of enantiomerically-enriched cyclopropyl ketones following deracemization by Al-salen photocatalysis. Reaction conditions: a) 1. Hydroxylamine hydrochloride, pyridine, EtOH, 85 °C, 16 h; 2. Tf₂O, DCM, r.t., 3 h; b) Me₃Si, NaH, DMSO, r.t., 16 h; c) LiAlH₄, THF, 0 °C to r.t., 16 h; d) TFAA, H₂O₂, DCM, 0 °C to r.t., 18 h; e) LiEt₃BH, THF, 0 °C to r.t., 4 h; f) LiAlH₄, THF, 0 °C to r.t., 18 h; e.r. determined after derivatization by chiral HPLC. For full experimental details see Supporting Information.

Reviewer Reports on the First Revision:

Referees' comments:

Referee #1 (Remarks to the Author):

I still have the question/point about CVs: if the CV is irreversible, you cannot get an accurate estimation of the redox potential of the substrates. Considering that the claimed difference between the two substrates is quite small (-0.16 V), I do wonder whether the uncertainty in the determination of the redox potentials that could be bigger than this?

The CVs in the presence of the catalyst does not show any changes – is this consistent with Lewis acid binding? Overall, it was not clear whether any of the analytical measurements (e.g. UV-vis) provided unambiguous support for this coordination.

Regarding the computed structures on page S94 showing (R)-CP1 and (S)-CP1 – it looks to me based on the geometries like they both would lead to a Z-configured intermediate upon C-C breaking? I couldn't reconcile this with the main text, where the opposite enantiomers result in the diastereomeric diradical intermediates.

The explanation of the computed PES has been substantially revised and additional details added. I think it's improved but I wanted to focus on a particular aspect of this, specifically: "(E)- and (Z)-3INT2 are energetically close to the (open-shell) singlet potential-energy surface, which would lead to facile intersystem crossing". In my opinion, however, this ISC would still need to be slower than the isomerization barrier between E/Z diradicals, so it can't be that fast?

Referee #2 (Remarks to the Author):

The resubmitted manuscript has appropriately addressed all of the concerns raised during the initial review period. The addition of more thorough DFT studies, product derivatizations, and control experiments (to the SI) has significantly strengthened the work. My earlier assessment - that this work is highly conceptually significant, certain to inspire significant additional investigation, and likely to appeal to a broad audience - continues to hold true. My view is that the amended manuscript is now suitable for publication in Nature.

Referee #3 (Remarks to the Author):

Response to comments:

The authors have undertaken a number of transformations to elaborate the enantioenriched products. This has demonstrated the utility of the products and is a helpful addition to the manuscript.

With relation to the trial of alternative substrates to highlight the reaction-mode generality,

including cyclobutanes and heterocycles, the authors have made a good effort to test these. Regrettably, these appear to either be synthetically intractable or not-effective. While it is disappointing these substrates are unsuccessful, it is clear that the an alternative asymmetric cyclization reported in relation to the other reviewers comments is encouraging and, in my view, sufficient to overcome this limitation in this manuscript.

The authors have attempted to prepare a number of alternative catalyst structures to evaluate the electronic effects. It is clear that the two tBu constituents are essential to provide steric barriers and so alternative and so additional substituents would be required in addition and, to this end, would be synthetically intractable and remove from the utility of the transformation. An alternative would be exploring the electronic effect of the diphenyldiamine backbone. These could however be explored in future manuscripts if further mechanistic exploration is to be conducted.

The authors argument against Cl fission, based upon the efficacy of the Al-F complex, is compelling.

Overall, I feel that the authors have adequately addressed my comments.

Author Rebuttals to First Revision:

Referee #1 (Remarks to the Author):

I still have the question/point about CVs: if the CV is irreversible, you cannot get an accurate estimation of the redox potential of the substrates. Considering that the claimed difference between the two substrates is quite small (-0.16 V), I do wonder whether the uncertainty in the determination of the redox potentials that could be bigger than this?

Author response. We appreciate the referee highlighting this point. We aligned our analysis with several leading studies in the literature, but fully appreciate that the *Supplementary Information* should be modified to reflect this uncertainty. The following statement has been included:

“Considering that the difference between the two substrates 1 and 19 is quite small (-0.16 V), care should be exercised when interpreting the results due to potential uncertainty in the determination.”

The CVs in the presence of the catalyst does not show any changes – is this consistent with Lewis acid binding? Overall, it was not clear whether any of the analytical measurements (e.g. UV-vis) provided unambiguous support for this coordination.

Author response. The referee is quite right that no changes were observed by CV in the ground state. The involvement of Lewis acid binding was part of the original working hypothesis but shown by our spectroscopic and computational investigations not to be operational. The DFT analysis indicates that Lewis acid binding only becomes relevant in the excited state. This clarification was requested by the referees in the previous revision and is described in detail (*please see the Supplementary Information*).

Regarding the computed structures on page S94 showing (R)-CP1 and (S)-CP1 – it looks to me based on the geometries like they both would lead to a Z-configured intermediate upon C-C breaking? I couldn't reconcile this with the main text, where the opposite enantiomers result in the diastereomeric diradical intermediates.

Author response. We appreciate the referee highlighting this important point. In the main text, we do indeed highlight this:

“The configuration of the double bond of the enol radical in 3INT2 depends on the cyclopropyl configuration in the low energy conformation of 3INT1: the (Z)-enol radical is formed from (S)-3INT1, whereas the ring opening of the (R)-ketyl radical furnishes the (E)-enol radical.”

We hope that this addresses the referee's concerns.

The explanation of the computed PES has been substantially revised and additional details added. I think it's improved but I wanted to focus on a particular aspect of this, specifically: “(E)- and (Z)-3INT2 are energetically close to the (open-shell) singlet potential-energy surface, which would lead to facile intersystem crossing”. In my opinion, however, this ISC would still need to be slower than the isomerization barrier between E/Z diradicals, so it can't be that fast?

Author response. This is an excellent point, but we respectfully believe that this is highlighted in the main text of the manuscript. We have, however, added an additional line to strengthen the point that this assumes the process are fast relative to the ISC rates.

“Fast conformational interconversions of diradicals (E/Z)-3INT2 can pre-select conformations which lead to the observed accumulation of the (S)-cyclopropane during bond formation in the singlet state, assuming that these processes are fast compared to ISC rates.”

We hope that this addresses the referee's concern.

Referee #2 (Remarks to the Author):

The resubmitted manuscript has appropriately addressed all of the concerns raised during the initial review period. The addition of more thorough DFT studies, product derivatizations, and control experiments (to the SI) has significantly strengthened the work. My earlier assessment - that this work is highly conceptually significant, certain to inspire significant additional investigation, and likely to appeal to a broad audience - continues to hold true. My view is that the amended manuscript is now suitable for publication in Nature.

Author response. No changes were requested by Referee #2.

Referee #3 (Remarks to the Author):

The authors have undertaken a number of transformations to elaborate the enantioenriched products. This has demonstrated the utility of the products and is a helpful addition to the manuscript.

With relation to the trial of alternative substrates to highlight the reaction-mode generality, including cyclobutanes and heterocycles, the authors have made a good effort to test these. Regrettably, these appear to either be synthetically intractable or not-effective. While it is disappointing these substrates are unsuccessful, it is clear that the an alternative asymmetric cyclization reported in relation to the other reviewers comments is encouraging and, in my view, sufficient to overcome this limitation in this manuscript.

The authors have attempted to prepare a number of alternative catalyst structures to evaluate the electronic effects. It is clear that the two tBu constituents are essential to provide steric barriers and so alternative and so additional substituents would be required in addition and, to this end, would be synthetically intractable and remove from the utility of the transformation. An alternative would be exploring the electronic effect of the diphenyldiamine backbone. These could however be explored in future manuscripts if further mechanistic exploration is to be conducted.

The authors argument against Cl fission, based upon the efficacy of the Al-F complex, is compelling.

Overall, I feel that the authors have adequately addressed my comments.

Author response. No changes were requested. However, we wish to note that the impact of the electronic effect of the diphenyldiamine backbone was explored and included in the previous revision. This information is provided in the *Supplementary Information*.